# Styrene Production in Genetically Engineered *Escherichia coli* in a Two-Phase Culture

**DOI:** 10.3390/biotech13010002

**Published:** 2024-01-14

**Authors:** Shuhei Noda, Ryosuke Fujiwara, Yutaro Mori, Mayumi Dainin, Tomokazu Shirai, Akihiko Kondo

**Affiliations:** 1Graduate School of Science, Technology and Innovation, Kobe University, 1-1 Rokkodai, Nada, Kobe 657-8501, Japan; akihiko.kondo@riken.jp; 2PRESTO, Japan Science and Technology Agency (JST), Kawaguchi 332-0012, Saitama, Japan; 3Center for Sustainable Resource Science, RIKEN, 1-7-22, Suehiro-cho, Tsurumi-ku, Yokohama 230-0045, Japan; ryosuke.fujiwara@riken.jp (R.F.); tomokazu.shirai@riken.jp (T.S.); 4Department of Chemical Science and Engineering, Graduate School of Engineering, Kobe University, 1-1 Rokkodai, Nada, Kobe 657-8501, Japan; yutaro.mori@hawk.kobe-u.ac.jp

**Keywords:** styrene, *trans*-cinnamic acid, *Escherichia coli*, two–phase culture, phenylalanine ammonia lyase, ferulic acid decarboxylase, oleyl alcohol, *Arabidopsis thaliana*, *Brachypodium distachyon*, pyruvate kinase

## Abstract

Styrene is an important industrial chemical. Although several studies have reported microbial styrene production, the amount of styrene produced in batch cultures can be increased. In this study, styrene was produced using genetically engineered *Escherichia coli*. First, we evaluated five types of phenylalanine ammonia lyases (PALs) from *Arabidopsis thaliana* (AtPAL) and *Brachypodium distachyon* (BdPAL) for their ability to produce *trans*-cinnamic acid (Cin), a styrene precursor. AtPAL2-expressing *E. coli* produced approximately 700 mg/L of Cin and we found that BdPALs could convert Cin into styrene. To assess styrene production, we constructed an *E. coli* strain that co-expressed AtPAL2 and ferulic acid decarboxylase from *Saccharomyces cerevisiae*. After a biphasic culture with oleyl alcohol, styrene production and yield from glucose were 3.1 g/L and 26.7% (mol/mol), respectively, which, to the best of our knowledge, are the highest values obtained in batch cultivation. Thus, this strain can be applied to the large–scale industrial production of styrene.

## 1. Introduction

The concept of carbon neutrality has gained worldwide recognition in the context of global warming and atmospheric pollution. The bulk production of chemicals and energy using environmentally friendly systems can open new avenues to promote a sustainable society [1]. Microbial fermentation is one of the most environmentally friendly processes and is widely used to produce chemicals, fuels, and pharmaceuticals [2,3,4,5,6,7,8]. Over the past two decades, tools for genetic engineering and strategies for metabolic engineering and synthetic biology have drastically advanced, and researchers have enabled their ideas to be realized freely in microbial metabolic pathways [5,9,10,11,12,13,14].

The production of volatile compounds by microbes has recently been studied extensively. For instance, isobutanol and isobutyl aldehydes produced by metabolically engineered *E. coli* were collected using a gas-stripping system [15,16]. Biphasic culture with an organic solvent is another approach for separating the volatile compounds of interest from the culture broth. Luo et al. [17] reported the production of methyl anthranilate, a grape-flavoring compound, using a two–phase culture system with tributyrin as an organic solvent. Furthermore, biphasic separation systems have great advantages in terms of toxicity against microorganisms, especially for toxic aromatic chemicals [18]. Biphasic cultivation can also be applied for the biotransformation of toxic compounds [19]. Luo et al. reported the production of methyl anthranilate and succeeded in the bioconversion of *p*-xylene into terephthalic acid in a two-phase cultivation with organic acid. To overcome various problems associated with the use of *p*-xylene as the substrate, such as volatility, insolubility, and microbial toxicity, oleyl alcohol (OA) was selected as the organic phase in the two–phase cultivation.

We previously reported the microbial production of 1,3-butadiene, a C4 unsaturated compound, from glucose, using *Escherichia coli* as the host strain [20]. The biosynthetic pathway of 1,3-butadiene was extended from the shikimate pathway via cis, cis-muconic acid. This chemical is industrially important as a raw material for styrene-butadiene-rubber (SBR), with a global market of $19 billion in 2019 [21]. Styrene and 1,3-butadiene are the key alternative compounds for SBR production. Thus, while the current process for SBR synthesis depends on chemical plants using fossil resources, the microbial production of 1,3-butadiene and styrene could facilitate a completely green process for the production of “Bio-SBR”.

Microbial styrene production has been widely studied by several research groups [22,23,24,25,26,27,28,29,30]. Using *E. coli* as the host strain, endogenous l-phenylalanine (Phe) is converted to styrene via *trans*-cinnamic acid (Cin) by the endogenous sequential reactions of L-phenylalanine ammonia lyase (PAL) and ferulic acid decarboxylase (FDC). Lee et al. [17] reported the production of styrene via the fed-batch cultivation of genetically modified *E. coli*. Although the optical density of the cells was greater than 100, the amount of styrene produced was approximately 5 g/L. In addition, to effectively collect the produced styrene, in situ gas–stripping recovery was adopted in that report. Together, these results indicate that the maximal level of styrene production in previous reports was achieved simultaneously by modifying the metabolic pathway and optimizing culture conditions. In other reports on styrene production using batch cultures, the styrene titer was below 1 g/L [22,23,24,25,26,28]. These results implied that styrene production per unit cell in *E. coli* requires further improvement.

In our previous studies, we developed a microbial platform to produce aromatic and derivative chemicals, including 1,3-butadiene, as mentioned above [20,31,32]. We hypothesized that our platform strain, CFT3, could be used for styrene production, similar to that demonstrated using other chemicals. To test this, we used our CFT3 strain, a Phe-overproducing *E. coli*, as the base strain and introduced a gene set encoding PAL from *Arabidopsis thaliana* (AtPAL) and FDC from *Saccharomyces cerevisiae*. To recover volatile styrene, we performed biphasic culture using OA. We successfully demonstrated styrene production using a two–phase batch cultivation process and obtained the highest reported amount of styrene production in microbial batch cultivation.

## 2. Materials and Methods

### 2.1. Strains and Plasmid Construction

The strains and plasmids used in this study are listed in Table 1. *E. coli* NovaBlue-competent cells (Novagen, Cambridge, MA, USA) were used for gene cloning. Polymerase chain reaction was performed using KOD-Multi & Epi-DNA polymerase (Toyobo, Osaka, Japan); the primer pairs used are listed below. Each gene was assembled with the respective plasmid using NEBuilder HiFi DNA Assembly Master Mix (New England Biolabs, Ipswich, MA, USA).

The plasmids pZE12-AtPAL2, pZE12-BdPAL1, pZE12-BdPAL2, pZE12-BdPAL6, and pZE12-BdPAL8 were constructed by cloning each synthetic gene fragment into the *Kpn*I site of pZE12-Ptrc.

To construct the plasmid pSAK-Ptrc, the Ptrc gene fragment was amplified by PCR using pZE12-Ptrc as the template and 5′-CCCTTTCGTCTTCACTGTTGACAATTAATCATCCG-3′ and 5′-AACCCGTACCCTAGGAAGGCCCAGTCTTTCGACTG-3′ as the primer pair. The plasmid fragment pSAK was amplified by PCR using pSAK as the template and 5′-GTGAAGACGAAAGGGCCTCG-3′ and 5′-CCTAGGGTACGGGTTTTGCT-3′ as the primer pair. The PCR fragments were conjugated, and the resulting plasmid was named pSAK-Ptrc.

To construct pSAK-FDC1, the FDC1 gene fragment was amplified by PCR using the synthetic gene fragment of FDC1 as the template and 5′-AAGGAGGAATAAACCATGCGTAAACTGAATCCGGC-3′ and 5′-TCTCGAGCTCGATCTTATTTATAGCCGTAGCGTT-3′ as the primer pair. The plasmid fragment pSAK-Ptrc was amplified by PCR using pSAK-Ptrc as the template and 5′-GATCCGAGCTCGAGATCTGC-3′ and 5′-GGTTTATTCCTCCTTATTTA-3′ as the primer pair. The PCR fragments were conjugated, and the resulting plasmid was named pSAK-FDC1. The sequence of each synthetic gene is summarized in the Appendix A.

Plasmids were transformed into bacterial strains using the Gene Pulser II (Bio-Rad, Hercules, CA, USA). We created five types of Cin-producing *E. coli:* CFT3A2, CFTB1, CFTB2, CFTB6, and CFTB8, expressing AtPAL, BdPAL1, BdPAL2, BdPAL6, and BdPAL8, respectively. To create styrene–producing *E. coli*, we introduced pSAK-FDC1 into the CFT3A2 strain, which was called CFT3A2FD.

Where applicable, 100 µg·mL^−1^ ampicillin and 15 µg·mL^−1^ chloramphenicol were added to the medium for selection. The transformants used in this study are listed in Table 1.

### 2.2. Culture Conditions

M9Y medium was used for styrene production in 5 mL test tube-scale cultures. M9Y minimal medium contains (per liter): 10 g glucose, 5 g yeast extract, 0.5 g NaCl, 17.1 g Na_2_HPO_4_·12H_2_O, 3 g KH_2_PO_4_, 2 g NH_4_Cl, 246 mg MgSO_4_·7H_2_O, 14.7 mg CaCl_2_·2H_2_O, 2.78 mg FeSO_4_·7H_2_O, 10 mg thiamine hydrochloride, 40 mg L-tyrosine, and 40 mg L-tryptophan (Tyr and Trp were included because CFT3 is auxotrophic for these amino acids). To culture the CFT3 derivative strains, approximately 10 mM sodium pyruvate was added to the medium to promote bacterial growth in the initial phase. Each preculture was seeded in 5 mL of M9Y medium in a test tube at an initial optical density at 600 nm (OD_600_) of 0.05. Tube–scale cultures were incubated at 30 or 37 °C in a shaker at 180 rpm. IPTG (0.1 mM) was added to the culture medium at an OD_600_ of 0.5. An equal volume of OA was added to the culture medium after 24 h of cultivation.

Product yield was calculated as yield (mol·mol^−1^) = [produced compound, mol]/[consumed glucose, mol].

### 2.3. Analytical Methods

Cell growth was monitored by measuring OD_600_ using a UVmini-1240 spectrophotometer (Shimadzu, Kyoto, Japan). The glucose concentration in the culture supernatant was measured using a Glucose CII test kit (Wako Pure Chemical Industries, Kyoto, Japan), following the manufacturer’s protocol.

### 2.4. Analytical Methods for Compounds

To estimate the amount of styrene produced, gas chromatography–mass spectrometry was performed using a GCMS-QP2010 Ultra instrument (Shimadzu) equipped with a DB-FFAP column (60 m, 0.25 mm internal diameter, 0.5 mm film thickness; Agilent Technologies, Santa Clara, CA, USA). Helium was used as the carrier gas at a flow rate of 2.1 mL/min. The injection volume was 1 μL with a split ratio of 1:10. The amount of isobutanol was quantified as follows: the oven temperature was initially maintained at 40 °C for 1 min, after which it was gradually raised to 195 °C at 10 °C/min, and further to 250 °C at 120 °C/min before finally being maintained at 250 °C for 3 min. The total running time was 20 min. The other settings were as follows: interface temperature, 250 °C; ion source temperature, 200 °C; and electron impact ionization, 70 eV. The amount of Phe produced was quantified as follows: the oven temperature was initially maintained at 150 °C for 5 min, after which it was raised to 300 °C at 10 °C/min and then held at 300 °C for 5 min. The total running time was 25 min. The other settings were as follows: interface temperature at 250 °C, ion-source temperature at 200 °C, and electron impact ionization at 70 eV. The dried residues of Phe were derivatized for 60 min at 80 °C in 30 µL of *N*-(tert-butyldimethylsilyl)-*N*-methyl-trifluoroacetamide and 30 µL of *N*,*N*-dimethylformamide prior to analysis. Cycloleucine was used as the internal standard.

The amount of Cin produced was analyzed using high–performance liquid chromatography (Shimadzu) with a 5C_18_-MS-II column (Nacalai Tesque, Kyoto, Japan). The culture supernatants were separated from the culture broth by centrifugation at 21,880× *g* for 20 min. The column was operated at 30 °C at a flow rate of 1.2 mL·min^−1^. A dual–solvent system comprising of solvent A (50 mM phosphate buffer, pH 2.5) and solvent B (acetonitrile) was used. The gradient was initiated at 100% A (0–3 min), gradually shifted to a 50:50 mixture of A and B (3–6 min), maintained between 6 and 7 min, and subsequently shifted to 100% A (7–13 min). Product concentrations were determined using an ultraviolet absorbance detector (SPD-20AV; Shimadzu) at 254 nm.

## 3. Results and Discussion

### 3.1. Screening for the Optimized Gene Encoding PAL

Cin is a key precursor in styrene production (Figure 1). Several studies have been conducted on microbial Cin production using various microorganisms as hosts. *E. coli* has been widely used for Cin production; however, the amount of Cin produced in batch cultures is usually below 1 g/L owing to its toxicity [34,35,36]. Cin is produced in plants and used as a starting compound for synthesizing various phenylpropanoids, such as lignins [37]. It is also industrially important and used as a building block to produce organic thin films [34,35,36,38]. Cin production has been investigated in several studies. Bang et al. [38] reported Cin production using genetically modified *E. coli*. In their report, overexpression of various genes involved in the shikimate pathway and *Streptomyces maritimus*–derived PAL in an *E. coli* strain resulted in a Cin production of approximately 700 mg/L. In this study, the CFT3 strain was used for Cin production. This *E. coli* strain is a versatile host for producing shikimate pathway derivatives and has been successfully used to produce approximately ten valuable compounds [20,31,32].

First, we constructed a Cin-producing *E. coli* strain. In addition to AtPAL2 from *A. thaliana*, we focused on several plant PALs from *Brachypodium distachyon*, a model plant for cereal crops such as barley and wheat [37]. Figure 2 shows the culture profiles of the Cin-producing strains. The maximum amount of Cin produced was 683 mg/L in the culture supernatant of the CFT3A2 strain after 96 h, whereas the amount of Phe produced reached its highest level after 48 h of cultivation.

In the present study, we introduced AtPAL2 into the CFT3 strain, a versatile *E. coli* strain for producing shikimate derivatives, and successfully demonstrated Cin production with approximately the same production titer as that reported by Bang et al. [38]. These results indicate that our CFT3 strain is useful for aromatic chemical production and can be genetically engineered to convert Cin into other valuable chemicals.

Similar to *A. thaliana*, several PAL genes are distributed in the genome of *B. distachyon*. However, their versatility and utility in bioproduction research remain unclear. In this study, considering the presence of active PALs in plant genomes, with the exception of *A. thaliana*, we demonstrated Cin production using *B. distachyon* PAL (BdPAL)-expressing *E. coli*. Although the highest production of Cin was confirmed using AtPAL2, Cin production using BdPALs-expressing *E. coli* was also successfully achieved. Currently, we are investigating the substrate specificity of BdPALs for producing other useful chemicals.

### 3.2. Styrene Production Using Single-Phase Batch Culture with E. coli Co-Expressing PAL and FDC

Microbial styrene production was achieved by generating microbes expressing both PAL and FDC. Here, we introduced FDC from *S. cerevisiae* into a CFT3A2 strain expressing AtPAL2, and constructed CFT3A2FD for styrene production. FDC from *S. cerevisiae* has been used in other studies on styrene and butadiene production. Figure 3 shows the culture profiles of CFT3A2FD at 30 and 37 °C. As shown in Figure 3A, the rate of glucose consumption cultured at 37 °C was higher than that at 30 °C, whereas the maximum level of cell growth cultured at 30 °C was greater than that at 37 °C. At 37 °C culture, although the Phe and Cin concentrations in the culture supernatant gradually increased, they decreased after 24 and 72 h of cultivation, respectively. However, styrene production was not observed in the culture supernatants.

For CFT3A2 expressing only AtPAL2, 1380 mg/L Phe after 48 h of cultivation, which was the maximum amount obtained, gradually decreased to 1100 mg/L after 96 h of cultivation (Figure 2). In contrast, with CFT3A2FD co–expressing AtPAL2 and FDC1, 1060 mg/L Phe after 24 h of cultivation drastically decreased to 180 mg/L after 96 h of cultivation (Figure 3). These results suggest that Phe was converted to Cin in the CFT3A2FD strain culture. Although the amount of Phe decreased to almost 700 mg/L between 24 and 72 h of cultivation, the Cin production at that time was only 100 mg/L. Cin produced from Phe was converted to styrene FDC, which was volatilized from the culture supernatant.

### 3.3. Two-Phase Batch Cultivation of Styrene-Producing E. coli Using OA

Two-phase cultivation is generally adopted to collect compounds of interest to produce volatile chemicals using microbes as biocatalysts, with OA being widely used as an extraction solvent [39,40,41,42].

First, we tested whether the styrene initially added to OA was volatilized. Figure 4 shows the time course of styrene concentration in the culture media without *E. coli* in two cases with different concentrations: 5 or 0.5 g/L. In the case of 5 or 0.5 g/L added styrene, 84.7% or 87.9% remained, respectively, after 72 h of incubation under aerobic conditions.

For styrene production and in situ extraction using OA, we cultured CFT3A2FD strain in a two–phase system containing M9Y medium and OA. In this study, to encourage initial cell growth, OA was added to the culture medium after 24 h of cultivation. Figure 5 shows the culture profile of CFT3A2FD at 37 °C. First, to prevent volatilization of styrene from the OA phase, two-phase styrene production was performed using a screw–capped test tube after 24 h of cultivation. In this case, aerobic culture was performed for 24 h. As shown in Figure 5B, the maximum amount of styrene detected in the OA phase was 0.31 g/L after 72 h of cultivation. However, approximately 1.0 g/L Phe remained in the aqueous phase after 96 h of cultivation. Cin accumulation was not observed in the culture medium. The two–phase styrene production was performed under aerobic conditions. As shown in Figure 5B, the maximum amount of styrene detected in the OA phase was 3.04 g/L after 96 h of cultivation. The Phe concentration, which reached its maximum level after 48 h of cultivation, gradually decreased to below 0.1 g/L in the aqueous phase after 96 h of cultivation. Meanwhile, the maximum level of Cin detected in the aqueous phase was 0.1 g/L after 72 h of cultivation, which decreased to below 0.1 g/L after 96 h of cultivation.

In a previous report on styrene production, fed–batch fermentation using an exterior extraction module produced approximately 5 g/L styrene [27]. In the batch culture of the constructed strain, the maximum amount of styrene produced was approximately 1.7 g/L. The engineered *E. coli* used in that study expressed PAL from *S. maritimus*, whereas we used AtPAL2 for tCA production. In addition, both pyruvate kinases (PykF and PykA) in our platform strain CFT3 were completely inactivated, which enhanced the intracellular PEP pool. These differences in phenotype may have affected the amount of styrene produced in batch culture. The benchmarks for styrene production in the batch cultures are summarized in Table 2. Here, we achieved a styrene production of 3.1 g/L. To the best of our knowledge, a 26.7% (mol/mol) yield from glucose is the highest reported value. The NST74 strain is generally available for styrene production in *E. coli*. Our parental strain, ATCC31882, had a genotype similar to that of NST74. However, the Pmax of our strain was higher than that of NST–derivative strains (Table 2). This could be attributed to the inactivation of pyruvate kinases, which enhances the carbon flux to the shikimate pathway. The observed production of 1.7 g/L styrene by YHP05, whose pyruvate kinases are partially inactivated, strongly supports this suggestion [27]. In addition, inactivation of the phosphotransferase system, combined with overexpression of the endogenous glucose uptake system, composed of galactose permease (GalP) and glucokinase (Glk), could be sufficient to encourage carbon flux to styrene via the shikimate pathway. Thus, the versatility of our CFT3 strain for Phe derivative production is clearly demonstrated.

Although we adopted a two–phase culture with organic acids, this strategy may be difficult to apply to large–scale production of styrene. In addition, styrene produced after 24 h of cultivation could not be recovered because oleyl alcohol was added to the medium after 24 h of cultivation.

## 4. Conclusions

In this study, we produced styrene using genetically engineered *E. coli* in a two–phase batch culture with OA. The Production of Cin, a precursor of the styrene biosynthesis pathway, was achieved using PAL genes from *A. thaliana* and *B. distachyon*. The expression of PAL from *A. thaliana* and FDC1 from *S. cerevisiae* in the CFT3 strain, with enhanced carbon flux to the shikimate pathway, resulted in styrene production. The maximum amount of styrene produced in the OA phase was 3.1 g/L, which was the highest titer reported for microbial styrene production in batch culture. Thus, our strain can be applied to the large–scale production of styrene for industrial production. Currently, we are developing an alternative method to two–phase culture for recovering volatile styrene with high yield, such as in situ gas stripping recovery and gas–phase direct polymerization of polystyrene. In our previous report, we successfully produced 1,3-butadiene using *E. coli* by extending the shikimate pathway. By combining the system for styrene production developed here with the production technology of 1,3-butadiene, a clean process to produce SBR, whose synthetic method completely depends on chemical processes, could be achieved in the near future.

## Figures and Tables

**Figure 1 biotech-13-00002-f001:**
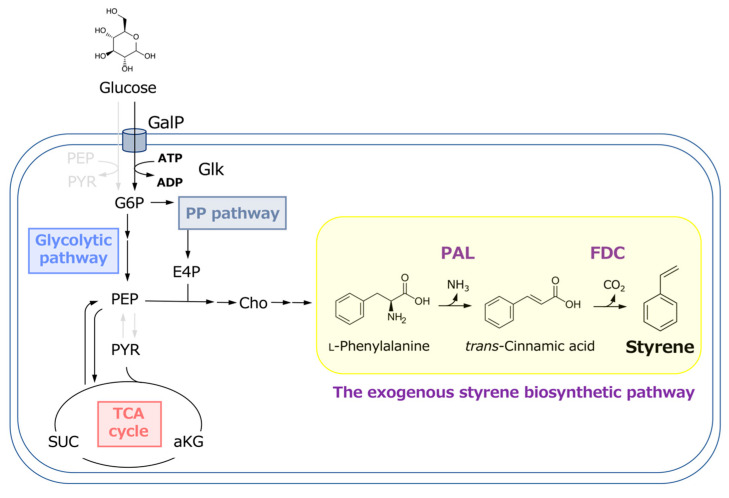
Metabolic pathway for styrene production from glucose in *E. coli*.

**Figure 2 biotech-13-00002-f002:**
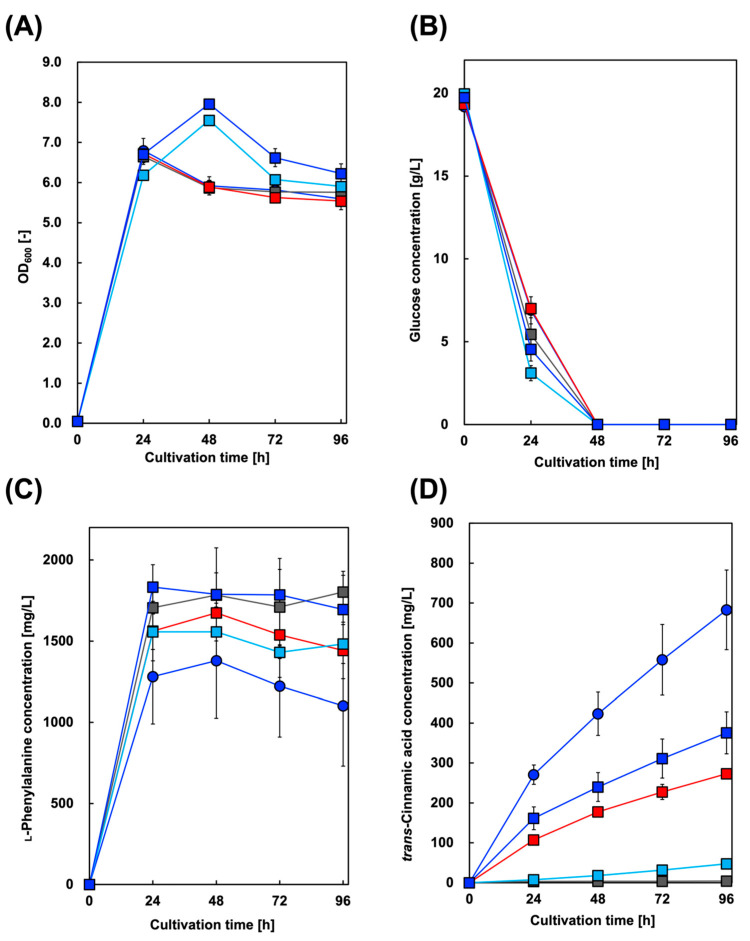
Culture profiles of *trans*-cinnamic acid-producing strains. Time course of (**A**) bacterial cell growth and (**B**) glucose consumption. The amounts of (**C**) l-phenylalanine and (**D**) *trans*-cinnamic acid produced in CFT3A2 (circles), CFT3B8 (blue squares), CFT3B2 (red squares), CFT3B6 (sky blue squares), and CFT3B1 (gray squares) cultures under aerobic conditions. Data are presented as the mean ± standard deviation of three independent experiments.

**Figure 3 biotech-13-00002-f003:**
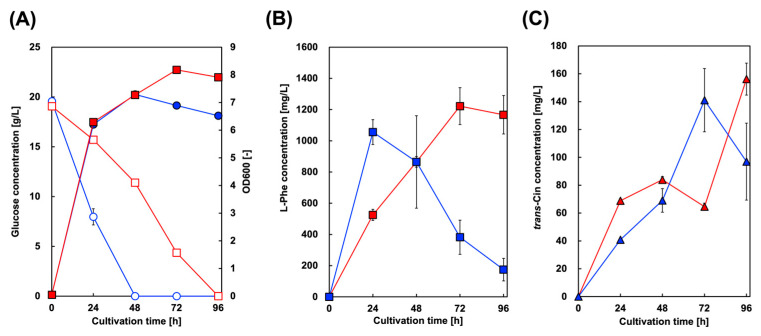
Culture profiles of styrene-producing strains in single-phase culture. Time course of (**A**) bacterial cell growth (closed symbols) and glucose consumption (open symbols) in CFT3A2FD cells. The amounts of (**B**) L-phenylalanine and (**C**) *trans*-cinnamic acid produced by CFT3A2FD at 30 °C (red symbols) and 37 °C (blue symbols). Data are presented as the mean ± standard deviation of three independent experiments.

**Figure 4 biotech-13-00002-f004:**
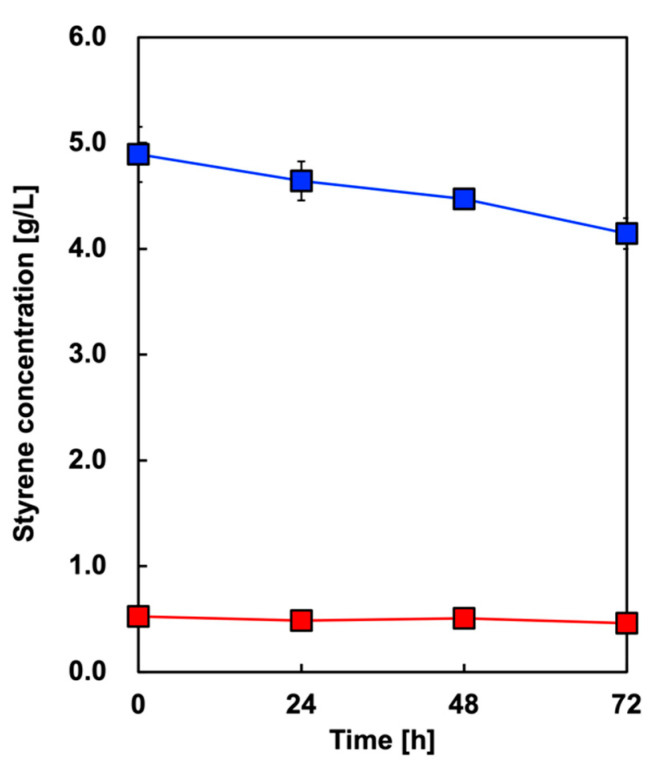
Time course of styrene concentration in the culture media without *E. coli* with 5 g/L (blue symbols) or 0.5 g/L (red symbols) of added styrene. Data are presented as the mean ± standard deviation of three independent experiments.

**Figure 5 biotech-13-00002-f005:**
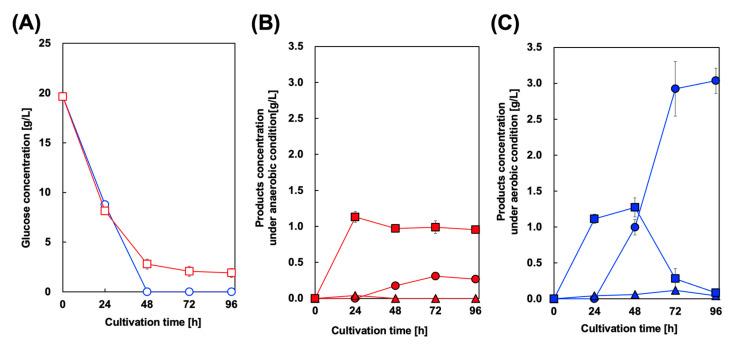
Culture profiles of styrene-producing strains in a two-phase culture. Time course of (**A**) glucose consumption in two–phase cultures of CFT3A2FD under aerobic (circles) and anaerobic conditions (squares). Time course of styrene (circles), *trans*–cinnamic acid (squares), and l-phenylalanine (triangles) concentrations under (**B**) anaerobic and (**C**) aerobic conditions. Data are presented as the mean ± standard deviation of three independent experiments.

**Table 1 biotech-13-00002-t001:** Strains, plasmids, and transformants were used in this study.

Strain or Plasmid	Genotype	Source or Reference
NovaBlue	*endA1 hsdR17*(rK12^−^mK12^+^) *supE44 thi-I gyrA96 relA1 lac* recA1/F’	Novagen
	[*proAB*^+^ *lacI*^q^ ZΔ*M15*::Tn*10*(Tet^r^)]; used for gene cloning	
ATCC 31882 (defined as CFT0)	L-phenylalanine-overproducing strain	American Type Culture Collection
CFT3	ATCC31882 *ptsHI*::*P_A1lacO-1_*-*Glk*-*GalP*Δ*pykF*Δ*pykA*	[31]
CFT3A2	CFT3 harboring pZE12-AtPAL2	This study
CFT3B1	CFT3 harboring pZE12-BdPAL1	This study
CFT3B2	CFT3 harboring pZE12-BdPAL2	This study
CFT3B6	CFT3 harboring pZE12-BdPAL6	This study
CFT3B8	CFT3 harboring pZE12-BdPAL8	This study
CFT3A2FD	CFT3A2 harboring pSAK-FDC1	This study
Plasmids		
pZE12-Ptrc	*P_trc_*, colE1 ori, Amp^r^	[33]
pSAK	*P_AlacO-1_*, SC101 ori, Cm^r^	[32]
pSAK-Ptrc	*P_trc_*, SC101 ori, Cm^r^	This study
pZE12-AtPAL2	pZE12-Ptrc containing *AtPAL2* from *A. thaliana*	This study
pZE12-BdPAL1	pZE12-Ptrc containing *BdPAL1* from *B. distachyon*	This study
pZE12-BdPAL2	pZE12-Ptrc containing *BdPAL2* from *B. distachyon*	This study
pZE12-BdPAL6	pZE12-Ptrc containing *BdPAL6* from *B. distachyon*	This study
pZE12-BdPAL8	pZE12-Ptrc containing *BdPAL8* from *B. distachyon*	This study
pSAK-FDC1	pSAK-Ptrc containing *FDC1* from *S. cerevisiae*	This study

**Table 2 biotech-13-00002-t002:** Benchmarks for styrene production using genetically engineered microbes.

Host	Genotype	Pmax (g L^−1^)	Yield (mol mol^−1^)	Substrate	Reference
*E. coli* CFT3	*aroG39*, *aroF394*, *PheA101*, *pheO352*, *tyrR366 tyrA4*, *trpE401*, *lacY5*, *malT384*, *thi-1 ptsHI::P_A1lacO-1_-glk-galP*Δ*pykF*Δ*pykA*	3.01	0.27	Glucose	This study
*E. coli* NST74	*aroH367*, *tyrR366*, *tna-2*, *lacY5*, *aroF394(fbr)*, *malT384*, *pheA101(fbr)*, *pheO352*, *aroG397(fbr)*	0.26	N.E.	Glucose	[22]
*S. cerevisiae* 22A75D	*MATa his3*Δ*0 leu2*Δ*0 met15*Δ*0 ura3*Δ*0 aro10*Δ*::aro4^K229L^*	0.029	0.0025	Glucose	[23]
*S. lividans* 1326 (coculture)	Wild-type strain	0.029	N.E.	Glucose	[24]
*E. coli* NST74	*aroH367*, *tyrR366*, *tna-2*, *lacY5*, *aroF394(fbr)*, *malT384*, *pheA101(fbr)*, *pheO352*, *aroG397(fbr)*	0.251	N.E.	Pyrolytic sugars	[25]
*E. coli* BL21(DE3)	*dcm ompT hsdS*(rB^−^ mB^−^) *gal*	0.35	N.E.	Glucose	[26]
*E. coli* YHP05	F^−^ l^−^ rph-1 INV(rrnD, rrnE)Δ*crr* Δ*tyrR* Δ*trpE* Δ*tyrA* Δ*pykA*	1.7	N.E.	Glucose	[27]
*E. coli* NST74	*aroH367*, *tyrR366*, *tna-2*, *lacY5*, *aroF394(fbr)*, *malT384*, *pheA101(fbr)*, *pheO352*, *aroG397(fbr)*	0.345	N.E.	Glucose	[28]

N.E.: Not estimated.

## Data Availability

Data are contained within the article.

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
