# Peer review of "Styrene Production in Genetically Engineered Escherichia coli in a Two-Phase Culture"

_biotech, 2024, doi:10.3390/biotech13010002_

Round 1

Reviewer 1 Report

Comments and Suggestions for Authors

This manuscript showed an engineered E. coli producing up to 3.1 g/L styrene, thus advancing microbial styrene production studies.

In order to have our future readers fully appreciate the significance of this study, I would suggest a more detailed comparison with the system used in reference 17. For example, how was the E. coli engineered?

Further along this line, do the authors plan to try fed-batch fermentation or continuous culture in view of Figure 5A? Are there any technical concerns? Also, does the TCA cycle under a regular condition of glucose inhibit styrene production in any way? If not, is there any experimental evidence?

Comments on the Quality of English Language

In addition, some very minor formality/language issues are noticed and detailed below.

(1) It seems the word “engineered” in the title does not need to be italic.

(2) Kindly select three keywords to present in lines 24-25.

(3) There seem two verbs in the sentence spanning lines 230 and 231.

Reviewer 2 Report

Comments and Suggestions for Authors

1. The hypothesis of the current investigation should be stated in the introduction section.

2. The rationale behind choosing specific strains and genetic modifications should be further elaborated upon. This would provide readers with a better understanding of why these particular modifications were necessary and how they contribute to enhanced styrene production.

3. The nature of replication in the experimental design is unclear, and the assessment of uncertainty in the reported measurement is absent or unclear. Information on the number of replicates conducted for each experiment and the variation observed would add credibility to the findings.

4. Table 2 lacks a more detailed comparative analysis with other existing methods or strains used for styrene production. This comparison is essential to highlight the novelty and efficiency of the proposed method.

5. The manuscript does not mention the analysis of by-products generated during the styrene production process. Understanding and managing by-products are crucial for the overall efficiency and environmental impact of the process.

6. There is no discussion of study limitations. This includes any difficulties in scaling up the process for industrial applications, potential by-products that might reduce the efficiency of the process, and how the strain might behave under different industrial conditions.

7. The discussion and suggestions for future work should also include comments on environmental impact and economic feasibility.

Comments on the Quality of English Language

Moderate edits required.

Round 2

Reviewer 1 Report

Comments and Suggestions for Authors

Many thanks for the authors’ patient elaboration. I support that the manuscript is ready for publication except for some super minor formality notes as detailed below.

(1)         It seems a hyphen is missing in lines 164 and 165.

(2)         The term “Escherichia coli” in line 57 is suggested to be changed to “E. coli” for consistency.

(3)         Shall we uncapitalize the word “Isobutanol” in line 43?

(4)         Also, shall we use “OA” instead of “oleyl-alcohol” in line 293 for consistency?

Reviewer 2 Report

Comments and Suggestions for Authors

Thank you for the revisions.

Comments on the Quality of English Language

Minor edits only.